# Localization and Laparoscopic Excision of Gastric Heterotopic Pancreas in a Child by Endoscopic SPOT^®^ Tattooing

**DOI:** 10.3390/children10020201

**Published:** 2023-01-22

**Authors:** Yu-Jung Liou, Shu-Chao Weng, Paul Chia-Yu Chang, Chuen-Bin Jiang, Hung-Chang Lee, Wai-Tao Chan, Cheng-Yu Ho, Pao-Shu Wu, Chun-Yan Yeung

**Affiliations:** 1Department of Pediatrics, Hsinchu Municipal MacKay Children’s Hospital, Hsinchu City 300046, Taiwan; 2Department of Pediatrics, MacKay Children’s Hospital, Taipei 104217, Taiwan; 3Division of Pediatric Surgery, Department of Surgery, MacKay Memorial Hospital, Taipei 104217, Taiwan; 4Department of Medicine, MacKay Medical College, New Taipei City 252005, Taiwan; 5MacKay Junior College of Medicine, Nursing, and Management, Taipei 112021, Taiwan; 6Division of Gastroenterology, Department of Internal Medicine, MacKay Memorial Hospital, New Taipei City 251020, Taiwan; 7Department of Pathology, MacKay Memorial Hospital, Taipei 104217, Taiwan

**Keywords:** endoscopic tattooing, children, heterotopic pancreas, laparoscopic surgery

## Abstract

Heterotopic pancreas (HP) is defined as pancreatic tissue lacking vascular or anatomic connection with the normal pancreas. Surgical resection is often indicated for symptomatic gastric HP. However, intraoperative identification of gastric HP is often difficult during laparoscopic surgery. Herein, we describe a patient with gastric HP, which was marked with SPOT^®^ dye (GI Supply, Camp Hill, PA, USA). The dye was seen clearly laparoscopically facilitating total excision of the lesion. The final pathology report confirmed the presence of heterotopic pancreatic tissue including pancreatic acini, small pancreatic ducts tissue with islets of Langerhans in the deep gastric submucosal area. There were no postoperative complications, and the patient was symptom-free. To the best of our knowledge, this was the first case report in the literature in which endoscopic tattooing of gastric HP before laparoscopic resection was performed. This method of localization was simple and reliable in children.

## 1. Introduction

Heterotopic pancreas (HP) is defined as pancreatic tissue lacking vascular or anatomic connection with the normal pancreas. In a prospective study, the prevalence of gastric HP was 1.1% in pediatric patients who underwent esophagogastroduodenoscopy (EGD) [1]. HP was symptomatic in 20% of patients in one study [2]. The most common presenting symptom was abdominal pain (23–50%) followed by nausea and vomiting (27%), weight loss (18%) and upper gastrointestinal hemorrhage (13%) [3]. Surgical excision is indicated if there is strong evidence of symptoms related to the lesion. The minimally invasive management for gastric HP varied between endoscopic submucosal dissection (ESD), endoscopic-guided laparoscopy and laparoscopic intraluminal (intragastric) surgery [4]. ESD is not widely practiced in children, and it is also more technically demanding. Endoscopic-guided laparoscopy is a less invasive management for gastric HP than laparoscopic intraluminal (intragastric) surgery.

However, intraoperative identification of gastric HP is often difficult during laparoscopic surgery. There are a limited number of case reports regarding the intraoperative localization of gastric HP in the literature. Christodoulidis et al. [5] reported an adult patient with gastric HP, which was marked with methylene blue during preoperative gastroscopy. Wedge resection of the lesion was performed successfully. Herein, we describe our technique of SPOT^®^ dye (GI Supply, Camp Hill, PA, USA) tattooing of gastric HP to facilitate laparoscopic resection. To our knowledge, this was the first case report in which we demonstrated that endoscopic tattooing for gastric HP before laparoscopic resection was simple and reliable in children.

## 2. Case Presentation

A 13-year-old boy presented to our hospital with a 3-month history of recurrent episodes of epigastric pain and a 1-week history of watery diarrhea with a frequency of 10 times a day, without blood or mucus. He had no nausea or vomiting but had weight loss of 4.2 kg within 2 months. Physical examination and plain abdominal X-rays were unremarkable and non-diagnostic. Laboratory studies including amylase, lipase and gastrin were normal. Stool culture and rotavirus antigen tests were negative. EGD revealed one oval subepithelial lesion with central umbilication located in the gastric antrum measuring approximately 10 mm in diameter (Figure 1A). The gastric mucosa appeared normal. HP was suspected. However, endoscopic conventional biopsy of the lesion showed chronic gastritis. Endoscopic ultrasonography revealed that the lesion was hypoechoic, heterogeneous with ill-defined margins, measuring 4.8 × 2.7 mm in size, arising from the submucosa (Figure 1B). A repeat endoscopic conventional biopsy of the lesion again showed chronic gastritis. Computed tomography of the abdomen revealed mild polypoid wall thickening, measuring 11 mm in diameter, located in the anterior wall of the gastric antrum. Laparoscopic resection of the lesion was arranged. Under general anesthesia, pediatric gastroscope (EG-760R, Fujifilm, Tokyo, Japan) was inserted through the mouth into the stomach. Endoscopic tattooing was performed by a pediatric gastroenterologist with a carbon-particle-containing solution, SPOT^®^, without dilution. The 23-gauge injection needle (Top Endoscopic Puncture Needle Digestive Tract, Top Surgical Taiwan Corporation, Kaohsiung, Taiwan) was punctured as perpendicularly as possible at four quadrants of the tumor deep into the muscle layer. Initially, 0.5 mL of SPOT^®^ was injected at the distal edge of the lesion. There was slight overflowing of the dye in the mucosa. This was followed by 0.3 mL of SPOT^®^ injected over the other 3 quadrants of the lesion (Figure 2A). Endoscopic tattooing was followed by laparoscopic resection of the lesion. The tattooing was seen near the antrum of the stomach (anterior surface) (Figure 2B). Gastrotomy was performed, and full thickness excision of the tumor was performed. The final pathology report confirmed the presence of heterotopic pancreatic tissue including pancreatic acini, small pancreatic ducts tissue with islets of Langerhans in the deep gastric submucosal area (Figure 3A,B). The patient was discharged 6 days after surgery without postoperative complications. He remained symptom free 1 year after surgery.

## 3. Discussion

Heterotopic pancreas was first reported by Jean-Schultz in 1727 when pancreatic tissue was found in an ileal diverticulum during the autopsy of a newborn [6]. It is also known as pancreatic heterotopia, ectopic pancreas, aberrant pancreas, accessory pancreas, or pancreatic rest. HP is defined as pancreatic tissue lacking vascular or anatomic connection with the normal pancreas. The frequency of HP on the autopsy specimens has been reported to range from 0.55% to 13.7% and can also be encountered incidentally during 0.9% of gastrectomies and 0.2% of upper abdominal surgeries [6,7]. The incidence of histologically verified HP in the pediatric population ranges from 6 to 16% [8]. In a prospective study, the prevalence of gastric HP was 1.1% in pediatric patients who underwent EGD [1]. HP can be found at different sites in the gastrointestinal tract including duodenum (27.7%), stomach (25.5%), jejunum (15.9%), diverticulum of gut (7.3%), Meckel’s diverticulum (5.3%) and ileum (2.8%) [9]. In the pediatric population, HP was found in Meckel’s diverticulum (26.6%), in the stomach, duodenum and jejunum (20%, respectively) and in the ileum (13.3%) [10]. It is less commonly reported in the colon, esophagus, omentum, spleen, liver or the mesentery. It is rarely found in the lungs, mediastinum and the gallbladder [3]. The most common location of gastric HP is in the greater curvature of the antrum [11,12]. Duodenal lesions are often found in the descending part of the duodenum, and jejunal lesions are mostly found at the level of the ligament of Treitz. Esophageal lesions are mainly discovered in the distal esophagus. Multiple sites of heterotopic pancreatic tissue can be found in the gastrointestinal tract. A 1-year-old girl was reported to have two HPs located on the duodenal wall and a third one in Meckel’s diverticulum [10]. In another case report, the sites of heterotopic pancreatic tissue were found in the stomach and duodenum [13].

The exact embryologic mechanism of HP is unknown, but several theories have been proposed. The misplacement theory suggests that fragments of pancreas become separated during rotation of foregut and develop into mature pancreatic tissue [11,14]. This theory may explain why HP is more commonly found in the proximal gastrointestinal tract. The metaplasia theory claims that endodermal tissues migrate to the submucosa and transform into pancreatic tissue [11,14]. The totipotent cell theory indicates that endodermal cells in gastrointestinal tract differentiate into pancreatic tissue [11].

LeCompte et al. [2] reported a retrospective study including 29 patients with HP in the upper gastrointestinal tract, and 6 patients (20%) were identified as having symptoms. HP was symptomatic in 14.1–100% of patients in other studies [3]. In a systemic review of 934 symptomatic patients identified with HP in the stomach or duodenum, the majority of patients presented with abdominal pain (67%) followed by dyspepsia (48%), pancreatitis (28%), gastrointestinal bleeding (9%) and gastric outlet obstruction (9%) [2]. The majority of the cases (90%) required surgical or endoscopic resection with 85% achieving resolution. Gastric HP in children may produce similar symptoms that include recurrent epigastric pain, vomiting from gastric outlet obstruction, gastrointestinal bleeding and diarrhea [15,16,17,18]. Clinical presentation of HP depends on several factors such as patient’s age, the location within the gastrointestinal tract, size of the lesion, mucosal involvement and timing of surgery. Older age, lesions greater than 1.5 cm in diameter, upper gastrointestinal tract location (stomach and duodenum) and mucosal layer involvement are more associated with the presence of symptoms [3]. Enzymes and hormones secreted by the heterotopic pancreatic tissue may explain symptoms presented in our patient [19]. A convincing causal relationship between gastric HP and its symptoms was established as symptoms resolved completely following surgery. Complications of HP include pancreatitis, gastrointestinal bleeding, bowel obstruction, intussusception, benign and malignant neoplasms and pseudocyst formation [11].

Imaging modalities for preoperative evaluation of gastric HP include gastrointestinal barium fluoroscopy, endoscopic ultrasonography, contrast-enhanced computed tomography and magnetic resonance imaging [12]. On barium study, gastric HP appears as an extramucosal intramural tumor, with a smooth surface and broad base [20,21]. Endoscopically, gastric HP has the typical appearance of an ill-defined, endoluminal, submucosal mass with central umbilication [7]. On endoscopic ultrasonography, gastric HP shows as a solid submucosal mass that is hypoechoic compared to the hyperechoic submucosa and isoechoic compared to the muscularis propria [22,23]. Computed tomography images show a small oval intramural mass with ill-defined or microlobulated margins and an endoluminal growth pattern [7,24]. On magnetic resonance sequences, gastric HP may follow the signal intensity of normal pancreas [22].

Definitive diagnosis of HP is made by histopathology. HP primarily arises in submucosa and can also be found in muscularis propria or subserosa [25]. In 1909, Heinrich proposed three types of HP. Type 1 HP tissue contains all the components of normal pancreatic tissue, including acini, ducts and islets of Langerhans; type 2 contains acini and ducts, with no islets of Langerhans; type 3 contains ducts only. Gasper-Fuentes in 1973 proposed a modified version of the Heinrich criteria. Type 1 HP tissue contains all the components of normal pancreatic tissue, including acini, ducts and islets of Langerhans; type 2 contains ducts only; type 3 contains acini only (exocrine); type 4 contains islets of Langerhans only (endocrine) [11]. Our patient is an example of a type 1 HP.

As mentioned previously, several characteristic endoscopic findings may help to establish a diagnosis of HP including typical localization in the gastric antrum, central umbilication and intact mucous membrane [19]. The differential diagnosis of gastric HP includes gastrointestinal stromal tumor, gastrointestinal autonomic nerve tumor, carcinoid tumor, lymphoma or gastric carcinoma [5]. Although our patient had characteristic features of gastric HP, diagnosis could not be confirmed by repeated biopsies since it was in the deep submucosal tissue. Total resection of the lesion, either endoscopically or laparoscopically, was considered the best means for diagnosis and treatment. The minimally invasive management for gastric HP varied between ESD, endoscopic-guided laparoscopy and laparoscopic intraluminal (intragastric) surgery. ESD is a less invasive alternative to surgery for patients with gastric HP. However, ESD is not widely practiced in children and is also more technically demanding. Laparoscopic intraluminal (intragastric) surgery was first described by Ohashi in Osaka and is an endoluminal surgery with guidance of the gastroscope [4]. Laparoscopic intraluminal (intragastric) surgery was considered to be too invasive for resection of gastric HP. Endoscopic-guided laparoscopy was a more attractive treatment option in our patient.

During laparoscopic gastric surgery, localization before surgical resection is crucial because lesions cannot be identified by inspection or palpation. Nine different methods have been reported for tumor localization during laparoscopic gastric surgery [26]. At present, three methods are commonly used. Endoscopic tattooing is most frequently used in the colon, although any area of the luminal gastrointestinal tract can be marked. Another method is endoscopic marking clip, which is used for localization in early gastric cancer [27,28]. The third method is endoscopy-assisted gastric resection during laparoscopic surgery [29]. Endoscopic dye tattooing was applied for localization in our patient.

SPOT^®^ (GI Supply, Camp Hill, PA, USA) is a specially formulated, biocompatible suspension designed for endoscopically tattooing lesions in the gastrointestinal tract [30]. The suspension contains highly purified, very fine carbon particles. It is the only U.S. Food and Drug Administration (FDA)-approved product for tattooing. Each steam sterilized, non-pyrogenic syringe contains 5 mL of water, polysorbate 80, glycerol, benzyl alcohol, simethicone and high-purity carbon black. It does not contain shellac, phenol, or ammonia present in India ink suspensions. Other substances for tattooing include India ink, indocyanine green, methylene blue, indigo carmine, hematoxylin, eosin, toluidine blue and isosulfan blue. SPOT^®^ for colonic lesion localization was still visible for up to 12 months in a previous study [31]. India ink and indocyanine green remained at the injection site after 2 days, whereas the other dyes were invisible within 1 day in an animal study [32]. We considered utilizing SPOT^®^ since its advantages include less inflammation than India ink and its safety and convenience in localizing gastric HP in children [33].

Hyman et al. proposed a technique of “four-quadrant” circumferential tattooing for effective colonic tattooing [34]. Wang et al. recommended the same technique for the intraoperative visualization of gastric subepithelial tumors in adults [33]. The technique requires the injection of SPOT^®^ at four quadrants of the tumors deep into the muscle layer, and the insertion of the needle perpendicularly as it is enough for visualization without spreading out. In our patient, the initial injection of 0.5 mL of SPOT^®^ at the distal edge of the lesion seemed slightly excessive. This was followed by 0.3 mL of SPOT^®^ injected at each of the other three quadrants of gastric HP. The use of 0.3 mL of SPOT^®^ was ideal for visualization during laparoscopic resection.

With regard to the optimal timing of endoscopic tattooing, Wang et al. [33] reported that there was no difference in surgical outcome whether the tattoo was done on the day of surgery or one day prior to surgery. If endoscopy was performed on the same day of surgery, the air inflation of the intestines could interfere with the visual field during laparoscopic surgery. However, repeated episodes of sedation/anesthesia exposure added an increased risk for children if the tattooing was done on the day before surgery. Studies have identified an association between exposure to anesthesia and neurodevelopmental deficit [35]. Knowing the negative effects of anesthetics on the developing brain, tattooing in our patient was performed on the same day of surgery in order to reduce preoperative anxiety, the frequency of the procedure and potential risk of adverse events of sedation/anesthesia.

There are a limited number of case reports about endoscopic tattooing for the excision of HP in the literature. Christodoulidis et al. [5] reported an adult patient with gastric HP, which was marked with methylene blue during preoperative gastroscopy. Laparotomy and wedge resection of the lesion was performed successfully. Chang et al. [36] also reported an adult patient with gastric HP on the posterior wall of the antrum, in which robotic partial gastrectomy was performed. In that case, endoscopic tattooing was performed for tumor localization before robotic surgery. Štor et al. [37] described an adult patient with gastric HP mimicking a gastrointestinal stromal tumor, which was tattooed proximally and distally before laparoscopic excision. Dhruv et al. [38] reported an adult patient with HP in the proximal jejunum, which was injected with indocyanine green dye during single balloon enteroscopy. Laparoscopic resection of the small bowel mass was performed successfully. To the best of our knowledge, this was the first case report in which endoscopic tattooing of gastric HP before laparoscopic resection was performed in a child.

There are some limitations in our case report. Firstly, although ESD was an option to accurately diagnose and treat gastric HP, we did not have sufficient experience in the pediatric population. Secondly, this was the first case report on endoscopic tattooing with SPOT^®^ dye before laparoscopic resection for gastric HP in a child. More experience in the pediatric population is needed to confirm its safety and efficacy.

## 4. Conclusions

In conclusion, to the best of our knowledge, this was the first case report in the literature in which endoscopic tattooing of gastric HP was performed just prior to laparoscopic resection under the same anesthesia. This method of localization was simple and reliable in children. More experience in the pediatric population is needed to confirm its safety and efficacy.

## Figures and Tables

**Figure 1 children-10-00201-f001:**
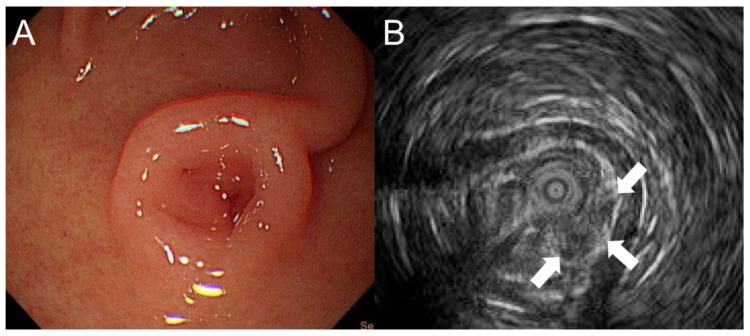
(**A**) Esophagogastroduodenoscopy revealing one oval subepithelial lesion with central umbilication located in the gastric antrum measuring approximately 10 mm in diameter. (**B**) Endoscopic ultrasonography showing that the lesion was hypoechoic, heterogeneous with ill-defined margins, measuring 4.8 × 2.7 mm in size, arising from the submucosa (arrow).

**Figure 2 children-10-00201-f002:**
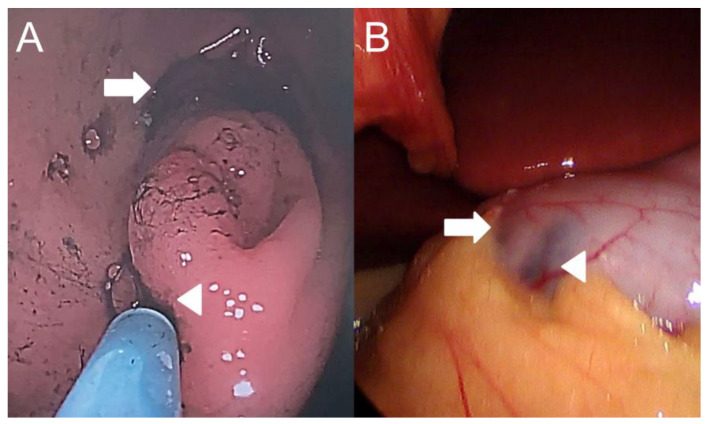
(**A**) Endoscopic view of tattooing with SPOT^®^ before laparoscopic resection of the lesion. The SPOT^®^ dye (GI Supply, Camp Hill, PA, USA) was clearly seen over the distal edge of the lesion after 0.5 mL injection. (**B**) Laparoscopic view of the tattooing in the antrum of the stomach (anterior surface). Arrow: distal side of the stomach in endoscopic and laparoscopic view. Arrowhead: proximal side of the stomach in endoscopic and laparoscopic view.

**Figure 3 children-10-00201-f003:**
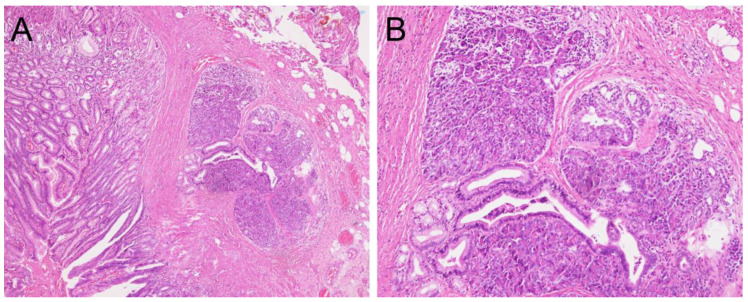
(**A**) Heterotopic pancreatic tissue in the deep gastric submucosal area (hematoxylin and eosin stain, 20×). (**B**) Enlarged image of heterotopic pancreatic tissue showing acini, small ducts with islets of Langerhans (hematoxylin and eosin stain, 100×).

## Data Availability

Not applicable.

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
