# Peer review of "Localization and Laparoscopic Excision of Gastric Heterotopic Pancreas in a Child by Endoscopic SPOT® Tattooing"

_children, 2023, doi:10.3390/children10020201_

Round 1

Reviewer 1 Report

Congratulations to the authors on this case report. Liou et al. described a case report of a heterotopic pancreas and presented their technique for tattooing the lesion before laparoscopic resection. 

I have some considerations: 

Title: It should be identified as a case report. Besides, it is hard to assume that the presented technique has safety and efficacy based on a non-analytical study.  

Introduction: The authors could split the paragraph in two.

Results: Are there more images of the surgery available? Are there images of the surgical specimen and histopathology slices?

Conclusion: It is hard to assume that the presented technique has safety and efficacy based on a non-analytical study.  

Reviewer 2 Report

The authors present a case with Gastric Heterotopic Pancreas in a 13-year-old boy. I only have a couple of comments:

It would be interesting to know how often Heterotopic Pancreas causes symptoms and/or requires an operation. Please add some data on this in the Introduction.

Please rephrase the conclusion. Although this case was treated successfully with this technique, we cannot conclude based on one case report only that this method is safe and reliable in children.
